# Improving Pea Protein Emulsifying Capacity by Glycosylation to Prepare High-Internal-Phase Emulsions

**DOI:** 10.3390/foods12040870

**Published:** 2023-02-17

**Authors:** Pere Morell, Adrián López-García, Isabel Hernando, Amparo Quiles

**Affiliations:** 1Facultad de Ciencias de la Salud, Universidad Internacional de Valencia—VIU, 46002 Valencia, Spain; 2Grupo de Microestructura y Química de Alimentos, Departamento de Tecnología de Alimentos, Universitat Politècnica de València, Camí de Vera, s/n, 46022 Valencia, Spain

**Keywords:** pea protein isolate, maltodextrin, glycosylation, HIPE

## Abstract

Pea protein has been extensively studied because of its high nutritional value, low allergenicity, environmental sustainability, and low cost. However, the use of pea protein in some food products is hindered due to the low functionality of pea protein, especially as an emulsifier. High-internal-phase emulsions (HIPEs) are attracting attention because of their potential application in the replacement of hydrogenated plastic fats in foods. In this study, the use of glycated pea protein isolate (PPI) as an emulsifier to prepare HIPEs is proposed. The functionalization of a commercial PPI in two ratios of maltodextrin (MD) (1:1 and 1:2) via glycosylation (15 and 30 min), to act as an emulsifier in HIPEs, is investigated. HIPE properties, such as oil loss and texture, were evaluated and related to microstructural properties. Glycated-PPI-stabilized HIPEs showed high consistency, firmness, viscosity, and cohesiveness values; a tight and homogeneous structure; and physical stability throughout storage. The results showed that emulsions were more stable when using a 1:2 ratio and 30 min of heat treatment. However, the reaction time was more determinant for improving the textural properties when a 1:1 ratio was used for glycosylation than when a 1:2 ratio was used. Glycosylation with MD via the Maillard reaction is a suitable method to enhance the emulsifying and stabilizing properties of PPI.

## 1. Introduction

Consumers are increasingly taking into account which components are used in the formulation of food products that they eat regularly [1]. The food industry has started to respond to increasing consumer demand for safe plant protein ingredients to replace those from animal sources [2,3]. Although animal proteins have been extensively researched and are commonly used in food products, projections of their adverse environmental impact are driving research on other protein sources, such as legumes. Legume proteins’ nutritional and functional properties (e.g., gelation, emulsification, and foaming) have led to their wide incorporation into food products due to substantial consumer acceptability, as well as their abundance and sustainability [4,5,6].

Pea protein has been extensively studied because of its high nutritional value, low allergenicity, environmental sustainability, and low cost [7,8]. However, use of pea protein in food products is hindered due to the low functionality of pea protein attributed to its protein structure and composition [9].

In a wide range of applications, emulsification is one of the most important functional properties of food proteins. However, the use of pea protein isolate (PPI) as a commercial emulsifier in food applications is still limited, which may be due to a poor performance in complex food systems or demanding production conditions and to the large performance gap between laboratory and commercially prepared pea proteins [10].

The emulsification properties of pea proteins are well characterized; however, they are sometimes evaluated by comparison with other protein sources, mainly soy protein, using the same emulsion system and the same measurement methods. Many studies have been performed comparing pea protein to some widely used reference emulsifiers. For example, PPI only outperformed soy protein isolate at lower oil concentrations (<10%), whereas at higher oil concentrations, PPI-prepared emulsions had larger droplets [11,12,13]. The inconsistent behavior of PPI could be due to its limited molecular flexibility, which could prevent it from making a stable interfacial film when more oil is present [14].

To overcome the functional limitations of the pea protein, various protein modification reactions have been studied. These include hydrolysis, succinylation, alkylation, phosphorylation, and glycosylation [9,15]. Many studies have reported that glycosylation of proteins and polysaccharides can modify the structure of protein and improve its functional properties, such as solubility, emulsification, and thermal stability [16,17,18]. This method shows promising potential, as the amino groups of proteins are covalently bonded to the carbonyl groups of the reducing sugars by the spontaneous Maillard reaction, with no other chemical reagents required for the reaction. Non-enzymatic glycosylation or glycation through the Maillard reaction can produce functional plant proteins without the addition of chemical products, and thus natural and clean-label food ingredients can be obtained [19]. In recent years, glycosylation modification and interaction between proteins and polysaccharides have attracted extensive attention from scholars worldwide because they are safe, green, and pollution free. Glycated products are often used as multifunctional additives due to their excellent properties [20].

The most important factors influencing the rate of the Maillard reaction are moisture, temperature, time, and pH [21]. Among these, temperature, time, and moisture are of great importance to be monitored during preparation. Therefore, the preparation techniques need to be designed to appropriately adjust the conditions. Maillard reaction is generally carried out using the dry-heating route [5]. However, this method works well at a laboratory scale, but the industrial application is complicated because the reaction needs prolonged time and is not easy to control. The wet-heating route, although less studied, can be also carried out to obtain conjugates via the Maillard reaction; this route improves the contact between the protein and the polysaccharide in the solution, decreasing the heating time and the time required for conjugation [22,23].

In the Maillard reaction, the creation of Amadori compounds is often used as an indicator of the first stage of the reaction. If the reaction continues, the Amadori compounds undergo several degradation reactions, and advanced glycation end products such as pentosidine and N(ε)-carboxymethyllysine are produced. In addition, the development of the brown color of “melanoidins” is used as an indicator of the advanced stages of the reaction. These end products can be unsafe for human well-being, leading to oxidative stress [24], and show also problems for industrial applications. The Maillard reaction needs to be controlled in the initial and intermediate stages to eliminate these products. As stated above, the use of the wet-heating route could reduce the heating time, and it has increasingly been used in the last few years [5]. However, the effect of this route to obtain conjugates on the functional properties of plant proteins has not been studied for short times of reaction. Therefore, to understand the relevance of glycosylation time, avoiding browning or melanoidin formation, in order to use the conjugates for industrial applications, we establish short glycosylation times, which are quite different from the longer times of reaction (more than 60 min) in the previous studies published.

Lately, high-internal-phase emulsion (HIPE) has shown remarkable potential in the food, pharmaceutical, and cosmetic industries. HIPEs are attracting increasing attention due to their potential application in the substitution of partially hydrogenated oils, encapsulating lipophilic functional ingredients, and processing porous materials and as catalyst supports and scaffolds for tissue engineering [25]. However, the production of such unique systems can be challenging because their excessive amount of oil fraction (>74%) could trigger several destabilization phenomena such as coalescence and phase separation [26]. Therefore, specific strategies must be applied to produce stable HIPEs. In this context, some protein–polysaccharide conjugates, such as whey protein–pectin and sodium caseinate–alginate, have been verified as surface-active agents in the stabilization of HIPEs [27,28,29]. These conjugates produced via covalent (Maillard-induced) interactions caused by the formation of irreversible and strong covalent links between amino groups of the proteins and carboxylic groups of polysaccharides have been used to obtain good stability in HIPEs [30]. Nevertheless, the consequence of glycosylation on the emulsifying properties of vegetal proteins in systems with high oil fractions is still unclear [31].

Up to now, glycosylation conjugates have been studied as emulsifiers in low or very low oil concentration emulsions. Furthermore, short glycosylation times have been established to avoid browning (melanoidin formation) to be used in industrial applications. In this study, we investigate the functionalization via glycosylation of a commercial PPI with maltodextrin (MD) to act as an emulsifier in significantly higher oil concentrations (HIPEs) and shorter times of reaction than research previously reported in the literature. To understand the relevance of glycosylation time and the protein–polysaccharide ratio involved, we assessed the properties of HIPEs, such as oil loss and texture properties, and related them to their microstructural properties.

## 2. Materials and Methods

### 2.1. Materials

Maltodextrin DE-19 was obtained from Sigma Aldrich (St. Louis, MO, USA). PPI (protein 84% dry basis) was supplied by the HSN Network Ltd. Eden Point (Granada, Spain). Sunflower oil was purchased from a local supermarket (Valencia, Spain).

### 2.2. Sample Preparation

#### 2.2.1. Synthesis of PPI–MD Conjugates via Glycosylation

PPI–MD conjugates were prepared by wet-heating route according to the method of Wen et al. [32] with minor modifications. After a preliminary screening, several ratios of PPI:MD were selected: 1:1 (sample a), 1:2 (sample b), and 1:0 (control). PPI and MD were added to distilled water and stirred for 1 h to ensure complete hydration. The pH of the medium was set to 9 using concentrated NaOH (6 M) to favor the Maillard reaction. At high pH values, the open-chain form of the sugar and the unprotonated form of the amino group, as reactive forms, favor the Maillard reaction [19]. The solution was introduced in 200 mL Erlenmeyer flasks and subjected to glycosylation via the wet-heating method in a temperature-controlled boiling water bath for 15 and 30 min at 90 ± 1 °C. These Maillard reaction times were used to avoid the formation of a brownish color of “melanoidins”, which indicates the advanced stages of the Maillard reaction [5] and can affect food applications. The solution was then immediately cooled to quench the reaction in an ice container. Finally, the sample was freeze-dried to obtain PPI–MD conjugates and stored at 4 °C. This produced four samples (15a, 15b, 30a, and 30b), and two samples heated without glycosylation were used as a control (15p and 30p).

#### 2.2.2. Preparation of High-Internal-Phase Emulsions (HIPEs)

HIPEs were produced using a food processor (TM31 Thermomix, Vorwerk, Wuppertal, Germany), using 1% of PPI–MD conjugate (glycosylated protein) or PPI (control) for stabilizing the different formulations. First, PPI–MD conjugate or PPI was dispersed in 100 g of water and sheared for 30 s at 300 rpm (speed 3). Next, 400 g of oil was progressively introduced into the processor via gravitation from a separatory funnel and mixed at speed 3 for 3 min. The speed was then increased to 1100 rpm (speed 4) and kept until all oil was added. Finally, the emulsion was sheared for 30 s at 3100 rpm (speed 6). Six HIPEs were obtained following this procedure: 15A, 15B, 15P, 30A, 30B, and 30P. All the samples were prepared at least three times.

### 2.3. Methodology

#### 2.3.1. Conjugate Solubility

The effect of glycosylation on protein solubility was evaluated according to the method of Zhang et al. [33]. PPI–MD conjugates were diluted to 10 mg/mL in phosphate-buffered saline (PBS, 1 M, pH 7). The solutions (1.5 mL) were added to Eppendorf tubes and centrifuged at 12,230× *g* for 20 min at low temperature (4 °C). The protein content of the supernatant was determined according to the Lowry method, measuring the absorbance at 280 nm and using bovine serum albumin as the standard. Protein solubility was expressed as soluble protein (protein concentration in the supernatant) per 100 g of protein (protein concentration in PPI–MD conjugates). Four replications were performed.

#### 2.3.2. HIPE Physical Stability

To study the physical stability of the emulsions, oil loss was analyzed using the method of Ye et al. [34] with some modifications, during 4 weeks at two storage temperatures: refrigeration temperature (4 ± 1 °C) and room temperature (20 ± 1 °C). Samples were introduced into 1.5 mL Eppendorf tubes and centrifuged for 30 min at 11,200× *g*. Then the free oil was carefully removed, and the oil loss value was obtained according to Equation (1), where *m*_0_ is the Eppendorf weight, *m*_1_ is the Eppendorf plus sample weight before centrifugation, and *m*_2_ is the Eppendorf plus sample weight after centrifugation. The assays were conducted in triplicate.
(1)$$Oil\;loss\;\left(\%\right)=\frac{m_{1}-m_{2}}{m_{1}-m_{0}}\times 100$$

#### 2.3.3. HIPE Textural Properties

The textural properties of the emulsions were determined according to the method of Liu, Xu, and Guo [35] with minor modifications. The tests were conducted using a TA-XT2 texture analyzer (Stable Micro Systems Ltd., Surrey, UK) equipped with a 30 kg load cell and a back-extrusion set comprising a compression plate (35 mm diameter) that was driven into a sample container (50 mm diameter and 70 mm height) to compress the sample. The measuring cup was filled with the sample, avoiding the formation of air bubbles, up to a height of 50 mm. The compression distance was 15 mm, the test speed was 1 mm/s, and the retraction speed was 5 mm/s. Parameters were obtained from the force–time curves using the TA.XTPlus Exponent software (Stable Micro Systems Ltd., Surrey, UK). The texture was determined the day after emulsion preparation (fresh samples), and after 2 and 4 weeks after storage at 4 °C. Texture measurements were performed at least three times.

#### 2.3.4. HIPE Microstructural Properties

The microstructure of the emulsions was analyzed by optical microscopy in bright-field mode using a Nikon Eclipse 80i optical microscope (Nikon Co., Ltd., Tokyo, Japan) with an ExWaveHAD camera, model no. DXC-190 (Sony Electronics Inc., Park Ridge, NJ, USA). The emulsions were carefully placed on a glass slide, and the images were captured and stored at 1280 × 1024 pixels using the microscope software (NIS-Element M, version 4.0, Nikon, Tokyo, Japan). Fresh HIPEs and HIPEs stored at 4 °C for 4 weeks were observed.

### 2.4. Statistical Analysis

Analysis of variance (one-way ANOVA) was performed on the data using Statgraphics Centurion XVI.II (StatPoint Technologies, Inc., Warrenton, VA, USA). Fisher’s least significant difference (LSD) test was used to evaluate the differences in mean values (*p* < 0.05).

## 3. Results and Discussion

### 3.1. Conjugate Solubility

Protein solubility is of paramount importance for any protein to perform its functional role, which is a prerequisite for other functional properties, such as emulsification [36]. Modifying the protein structure affects its properties, and therefore we evaluated the impact of wet glycosylation on the solubility of PPI at pH 7. This pH was chosen to avoid the range 4–6, which is the isoelectric point (pI) of the pea protein.

Glycosylation with MD produced a positive impact on protein solubility; significant differences (*p* < 0.05) were observed between the solubility of PPI–MD conjugates and that of the samples without MD (15a and 15b vs. 15p; 30a and 30b vs. 30p) (Figure 1).

When comparing the effect of the PPI:MD ratio, 15b and 30b had higher solubility (*p* < 0.05) than 15a and 30a. Protein solubility often depends on the balance between hydrophobic–hydrophilic groups on the protein surfaces [37]. The enhanced protein solubility of the samples with higher MD ratios can be attributed to the introduction of hydrophilic saccharide groups and their steric hindrance, which restrained the protein–protein interactions. Moreover, when proteins glycosylate with MD molecules, they can become more polar, which could make the conjugate a more soluble compound. The choice of polysaccharide and the degree of crosslinking are the critical factors in determining the physicochemical properties of the resulting protein–polysaccharide conjugates [38]. As stated by other authors, the structural characteristics of saccharides, in our case MD, and the number of saccharides attached to protein synergistically modulate protein functionalities [39].

When comparing the effect of the reaction time, 30a and 30b were more soluble (*p* < 0.05) than 15a and 15b. However, the first 15 min of the Maillard reaction was sufficient for conjugation between the protein and the polysaccharide via the formation of covalent bonds, which improved functional properties. These first minutes involve the condensation of the carbonyl group of the reducing sugar with the available ε-amino groups of the protein, resulting in Amadori products through the formation of a Schiff base [40]. This agrees with the results of C. Zhao et al. [31], who found good solubility properties in soy protein isolate–MD conjugates prepared with 20 min heat treatment.

Overall, glycosylation improved the solubility of PPI; the longer the reaction time and the greater the availability of the polysaccharide, the better solubility was achieved. Both the PPI:MD reaction time and the ratio significantly affected the solubility (*p* < 0.05). Therefore, glycosylation via the Maillard reaction is a feasible method to enhance the solubility of the pea protein.

### 3.2. HIPE Physical Stability

The physical stability of the HIPEs in terms of oil loss was measured during storage at 4 °C and 20 °C for 4 weeks. The storage stability of the emulsion is an important factor in determining its application in industrial production.

Figure 2A indicates the oil loss values of the HIPEs stored at 4 °C. Samples were compared for the same day of storage. During the first 2 weeks of storage (week 0 and week 1), there was no oil loss from the samples. HIPE samples showed great stability at refrigeration temperature during the first two weeks while maintaining their structure without oil loss.

After the third week of storage, only 15P showed a significantly higher oil loss than the rest of the samples (*p* < 0.05). This agrees with the results of Tirgarian et al. [30], who showed that HIPEs prepared using conjugates exhibited better stability compared to HIPEs stabilized with native proteins (soy protein isolate and sodium casein) when stored at 4 °C. They attributed these results to the combined emulsifying features of both biopolymers in the protein–polysaccharide conjugates and to the increased surface activity provided by polysaccharides. No significant differences were revealed in the oil loss measurements (Figure 2A) among the rest of the samples.

As expected, oil release started earlier when the samples were stored at 20 °C (Figure 2B). In the first week, PPI-stabilized HIPEs (15P and 30P) showed significantly higher oil loss (*p* < 0.05). No differences were found among the rest of the samples. With 2 weeks of storage, 15A, 30A, and 30B showed significantly less oil loss than the rest of the samples (*p* < 0.05). PPI-stabilized HIPEs (15P and 30P) stored at 20 °C lost over 30% of the total mass, showing they were the most unstable samples.

After 3 weeks, two trends were seen. PPI-stabilized HIPEs (15P and 30P) and 15B showed higher oil loss values. However, 15A, 30A, and 30B maintained the lowest values, showing good physical stability. Chen et al. [41] also observed a similar trend in O/W emulsions stabilized using whey protein isolate (WPI) and WPI–gum acacia (GA) conjugates produced by the Maillard reaction. These authors reported a significant improvement in the emulsion stability using WPI–GA conjugates as compared with WPI, which they attributed to a better solubility of the conjugates when compared to the emulsions stabilized with WPI. In PPI–MD stabilized emulsions with low oil content (10%), Zhang, Wang, and Adhikari [5] suggested that the improved stability could be attributed to the decrease in O/W emulsion droplet size due to the increased repulsive electrostatic force of the droplets and the improved steric hindrance offered by the PPI–MD conjugates.

### 3.3. HIPE Textural Properties

The back-extrusion test comprises applying a force to a material until it flows. Figure 3 indicates the force–time curves obtained in the back-extrusion experiments for the different fresh HIPEs. Similar texture profiles were observed between the samples, and all the force–time curves were smooth in the positive range. The glycated-PPI-stabilized samples (15A, 15B, 30A, and 30B) showed higher force values than the PPI-stabilized samples (15P and 30P). This might indicate a more elastic behavior in these emulsions. These characteristics are expected to occur in shortenings or margarines, as they are designed to easily flow or to spread under refrigeration conditions [42].

Texture characterization values are shown in Table 1. For fresh samples, 15B, 30A, and 30B showed the highest values (*p* < 0.05) for consistency and firmness parameters. The consistency value obtained by back extrusion provides a reference on the homogeneity of the product when a force is applied. The increased glycosylation time and MD proportion increase the consistency and firmness, leading to more homogeneous samples.

The cohesiveness parameter allows the evaluation of how cohesive or full-bodied the emulsion is, as a function of the time of application of the force capable of disintegrating the intermolecular forces that maintain the physical integrity of the emulsion [43]. Sample 30A showed the highest values (*p* < 0.05) in cohesiveness and viscosity index. Samples with higher consistency and viscosity index values impose resistance to the movement of the internal phase of the emulsion and avoid loss of stability [44].

Sample 15P measured at 24 h showed the lowest values for all texture parameters, followed by sample 30P. These results confirmed that PPI-stabilized HIPEs were the less homogeneous and cohesive samples.

After 2 weeks of storage, the consistency, cohesiveness, and firmness values of 15P and 30P increased significantly (*p* < 0.05), whereas those of the glycated-PPI-stabilized samples progressively decreased throughout storage (Table 1). Regarding the viscosity index parameter, all samples exhibited significant decreases (*p* < 0.05), except samples 15A and 15P. Other authors such as Fuhrmann, Sala, Stieger, and Scholten [45] have shown that the increase in viscosity can be an effect of the increase in the effective oil volume fraction as a result of the aggregation of the oil droplets; the larger the clusters of oil droplets in a given absolute volume fraction of oil, the larger the effective volume fraction of the droplets due to the entrapment of the bulk liquid between the oil droplets within the clusters.

After 4 weeks of storage, all measured values for samples 15A, 15B, 30A, and 30B decreased, and differences among them were reduced. These four samples showed significantly higher values in firmness and cohesiveness than 15P and 30P (*p* < 0.05), showing that glycated-PPI-stabilized HIPEs maintained good physical integrity after storage. However, PPI-stabilized HIPEs suffered the sharpest decline in values, showing their diminished textural characteristics throughout storage. Sample 30B showed the highest values for each texture parameter measured after 4 weeks of storage.

When the reaction time was 15 min, the PPI:MD ratio of 1:2 produced an increase in the values of the textural parameters, except for firmness. However, when the reaction time was 30 min, the change in the PPI:MD ratio had little influence.

Overall, the reaction time helps improve the textural properties when a 1:1 PPI:MD ratio is used for glycosylation. Glycosylation gave adequate textural properties in these emulsion formulations and the maintenance of them throughout storage.

### 3.4. HIPE Microstructural Properties

To reveal the underlying mechanism for the improvement of HIPE emulsion formation by glycated PPI, the microstructure of the HIPEs stabilized by PPI and glycated PPI (24 h and 4 weeks) was evaluated by optical microscopy (Figure 4).

Oil droplets in HIPEs often have polyhedral geometries because they are tightly squeezed together when the highest geometric limit for the packing of rigid spheres is exceeded [46]. All glycated-PPI-stabilized samples (15A, 15B, 30A, and 30B) observed after 24 h showed a tighter and more homogeneous distribution of oil droplets than PPI-stabilized samples (15P and 30P).

Glycated-PPI-stabilized samples with 30 min of heat treatment had smaller and better packing of oil droplets than samples with 15 min. Among 30 min glycated-PPI-stabilized samples, sample 30A showed a less structured system than sample 30B. The glycosylation time influenced the microstructure of HIPEs.

The B samples had better packing than the A samples. The protein–polysaccharide ratio also seemed to influence the microstructure of the emulsion samples. Other authors [47] have attributed the good emulsion-stabilizing properties of Maillard conjugates to a combination of the protein adsorption capacity and the high hydrophilicity of the carbohydrates, which would be giving steric stabilization to the oil droplets in the emulsion. Improving protein solubility by glycosylation led to more structured and stable systems.

After 4 weeks of storage, samples showed a wide droplet size distribution. The smaller droplets appeared between larger droplets, which is a microstructural feature of a highly concentrated emulsion comprising partially flocculated droplets [48]. Glycated-PPI-stabilized HIPEs better maintained their structure, with packed globules, reflecting the strengthening of their internal structural integrity. However, PPI-stabilized samples presented an unstructured and free oil phase, which can be related to a coalescence phenomenon. Although there is a high concentration of protein in this system, when the protein is not glycated, a “surfactant-poor” regime of emulsification could be established. Therefore, the initial globules, which were not completely coated by the protein, coalesce, and then their surfaces become fully coated by the protein molecules [49]. These results agreed with the textural results at 4 weeks when PPI-stabilized HIPE samples showed the smallest values of cohesiveness.

## 4. Conclusions

The solubility of pea protein isolate was enhanced when it was glycated with maltodextrin via Maillard reaction using the wet-heating route, which is attributed to the introduction of hydrophilic groups and their steric hindrance, limiting the protein–protein interactions. The obtained conjugates, specifically those prepared during 30 min of reaction time, provided excellent texture and structural stability to the concentrated emulsions. Therefore, short times of glycosylation with maltodextrin can be used to enhance the emulsifying and stabilizing properties of PPI in complex systems such as HIPEs. This study opens a new pathway to decrease the reaction time for the obtention of Maillard conjugates with good emulsifying properties, avoiding browning, which is an important drawback for industrial applications.

## Figures and Tables

**Figure 1 foods-12-00870-f001:**
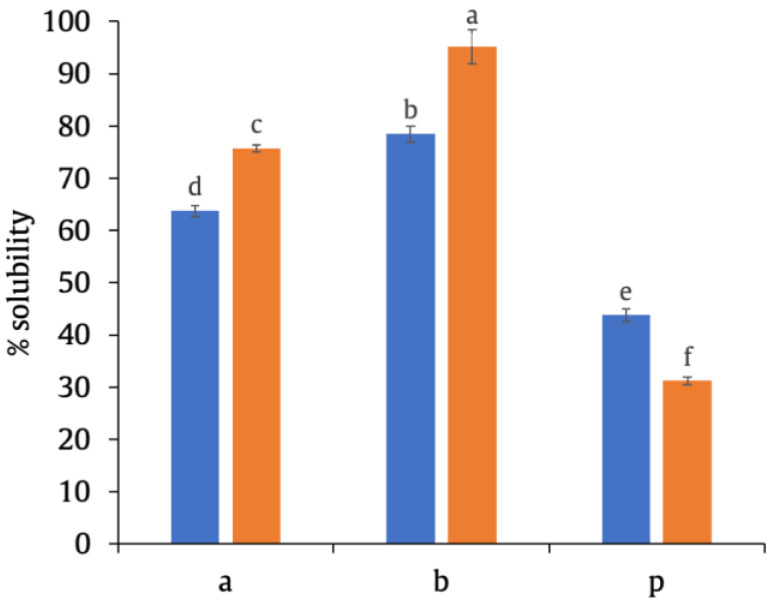
Effect of PPI:MD ratio and heat treatment time on the solubility of PPI–MD conjugates. a (PPI:MD ratio 1:1), b (PPI:MD ratio 1:2), p (without MD). Glycosylation time 15 min (●) and 30 min (●). Different letters indicate significant (*p* < 0.05) differences among samples according to the LSD multiple range test.

**Figure 2 foods-12-00870-f002:**
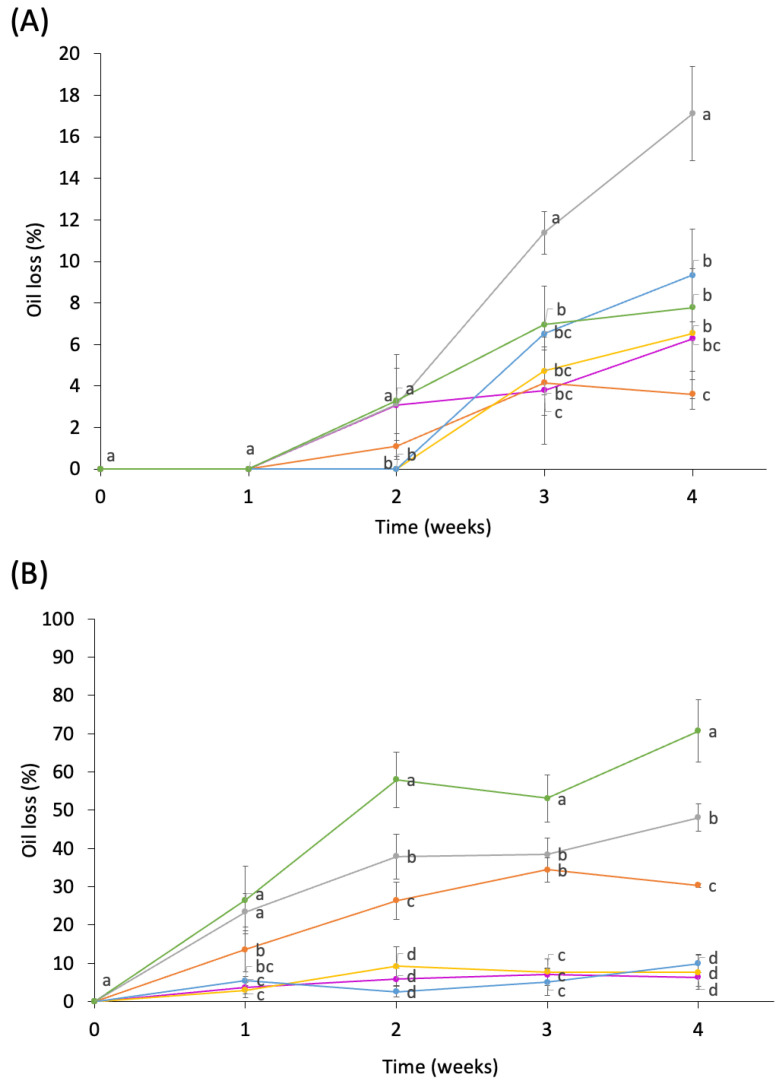
Oil loss (%) of samples stored at 4 °C (**A**) and 20 °C (**B**). 15A (●), 15B (●), 15P (●), 30A (●), 30B (●), and 30P (●). 15 (heated for 15 min), 30 (heated for 30 min). Different letters indicate significant (*p* < 0.05) differences among samples for the same day of storage according to the LSD multiple range test.

**Figure 3 foods-12-00870-f003:**
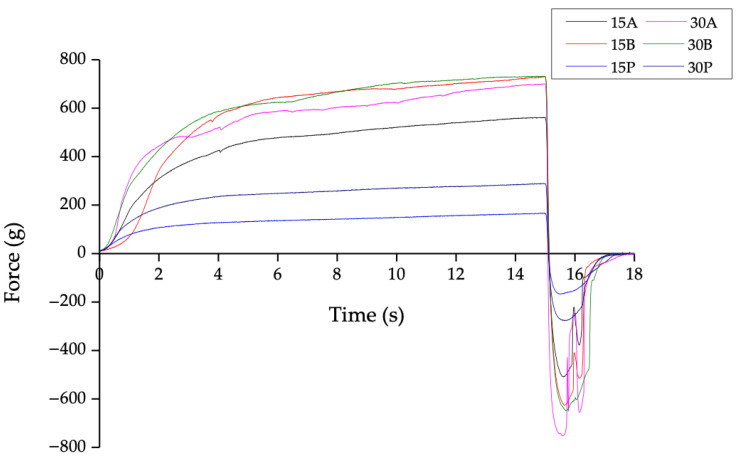
Back-extrusion force–time curves of fresh samples.

**Figure 4 foods-12-00870-f004:**
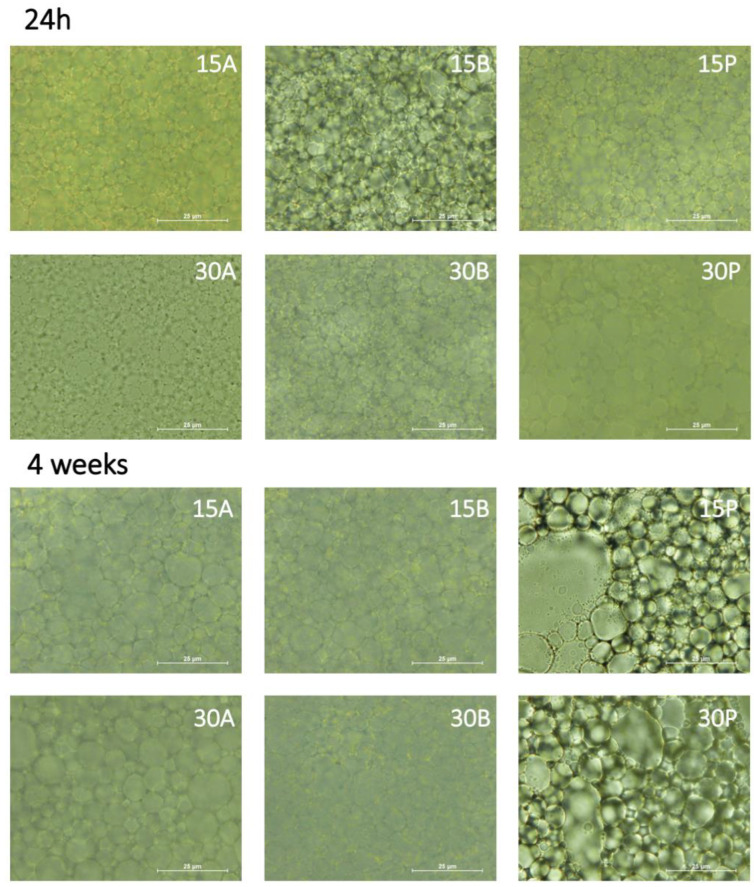
Light microscopy micrographs of fresh and 4 weeks stored (4 °C) HIPEs stabilized with PPI:MD conjugates.

**Table 1 foods-12-00870-t001:** Back-extrusion parameters of HIPEs at different refrigeration storage times.

Textural Parameters	Sample	Storage Time		
		Fresh	2 Weeks	4 Weeks
Consistency (g·s)	15A	6442 ± 262 ^bA^	6019 ± 203 ^bB^	2324 ± 139 ^cC^
	15B	8781 ± 231 ^aA^	4888 ± 366 ^dB^	3411 ± 341 ^aB^
	15P	2313 ± 287 ^dB^	4979 ± 312 ^cdA^	1445 ± 263 ^dC^
	30A	8553 ± 335 ^aA^	6734 ± 126 ^aB^	2671 ± 276 ^bC^
	30B	8713 ± 403 ^aA^	5334 ± 222 ^cB^	3098 ± 39 ^aC^
	30P	3647 ± 251 ^cB^	4209 ± 272 ^eA^	2019 ± 77 ^cC^
Cohesiveness (g)	15A	479 ± 25 ^cA^	460 ± 50 ^bA^	240 ± 21 ^aB^
	15B	649 ± 29 ^bA^	428 ± 37 ^bcB^	258 ± 30 ^aC^
	15P	171 ± 15 ^eB^	430 ± 2 ^bcA^	150 ± 40 ^bB^
	30A	760 ± 25 ^aA^	565 ± 35 ^aB^	259 ± 27 ^aC^
	30B	654 ± 11 ^bA^	460 ± 21 ^bB^	265 ± 19 ^aC^
	30P	290 ± 26 ^dB^	406 ± 32 ^cA^	176 ± 12 ^bC^
Firmness (g)	15A	559 ± 28 ^bA^	442 ± 54 ^bB^	260 ± 39 ^aC^
	15B	734 ± 19 ^aA^	406 ± 20 ^bB^	264 ± 29 ^aC^
	15P	174 ± 16 ^dB^	429 ± 22 ^bA^	125 ± 10 ^cC^
	30A	716 ± 35 ^aA^	568 ± 31 ^aB^	260 ± 21 ^aC^
	30B	739 ± 28 ^aA^	531 ± 34 ^aB^	275 ± 29 ^aC^
	30P	299 ± 26 ^cB^	348 ± 24 ^cA^	171 ± 6 ^bC^
Viscosity index (g·s)	15A	514 ± 17 ^cB^	656 ± 36 ^aA^	264 ± 46 ^bcC^
	15B	642 ± 38 ^bA^	485 ± 108 ^bB^	165 ± 119 ^dC^
	15P	225 ± 23 ^dB^	502 ± 19 ^bA^	191 ± 14 ^dB^
	30A	807 ± 70 ^aA^	261 ± 18 ^cB^	308 ± 7 ^aB^
	30B	672 ± 31 ^bA^	515 ± 23 ^bB^	291 ± 3 ^aC^
	30P	234 ± 5 ^dA^	166 ± 30 ^dB^	223 ± 12 ^cdA^

Values with different lowercase letters (a, b, …, z) within the same column are significantly different (*p* < 0.05) according to the LSD multiple range test for each parameter. Values with different capital letters (A, B, …, Z) within the same row are significantly different (*p* < 0.05) according to the LSD multiple range test for each parameter.

## Data Availability

Research data are not shared.

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
