# Peer review of "Improving Pea Protein Emulsifying Capacity by Glycosylation to Prepare High-Internal-Phase Emulsions"

_foods, 2023, doi:10.3390/foods12040870_

Round 1
Reviewer 1 Report
Reviewer Comments
This manuscript describes the effect of glycolylation of pea protein isolate with maltodextrin in their technological properties, such as emulsifying and textural properties to be used in HIPEs. The experimental design of the project is robust, and the authors describe in great detail their methods. Further work on the statistical analysis of the results could be done to obtain a better understanding of the effect of each variable studied. The discussion of the results increase our understanding of the potential of this glycosylated proteins and their applications in food products. Authors are encourage to do some revision of the work proposed in the lines below:
L20. ‘..good structure…’ is a very subjective way to describe a property. Authors should describe this quality in scientific terms.
L21. Replace ‘… more stable emulsions…’ with ‘showed that emulsions were more stable when…’
L30. Authors suggest that safety is one of the main concern of consumers that demand alternative protein sources. Authors should support this statement with some figures and references.
L32. Replace ‘food systems’with ‘food products’.
L32. When mentioning ‘adverse impact’ what type of impact are authors referring to? Please describe or specify.
L35. Replace ‘with’ with ‘due to’.
L67-68. This sentence contradicts the statement in lines 56-58. Please, rewrite in a consistent way.
L78. This paragraph could be a continuation of the previous one.
L103. Did the authors record/control the temperature of the solutions during reaction time? Was the temperature reach at two different times different? Could this have an impact on the type of reactions and chemical pathways dominating the reactions?
L104-105. How was the formation of browning/melanoidins controlled and checked? Were colour changes measured? Or melanoidins components analysed in the final products?
L119-L157Through the methods section it is not clear how many times the different samples were prepared, how many replicates per sample was produce and how many analytical replicates were run per analysis. Please, be more specific in each of the material and methods sections.
L140. Stable Micro Systems are based at Surrey.
L158-161. A better description of the statistical analysis done to assess each property analysed should be done. It is not clear if a one-way anova was performed for all the properties, or if for some of them results were compared by day, or by temperature. Authors should reconsider the statistical analysis performed as several variables (time, temperature, concentration M (0, a and b)) were part of the experimental design and their individual effect and/or interactions should be studied.
L181-182. Rewrite this sentence. Authors should explain in more detail how non-polar amino groups became polar.
L183. Replace ‘were’ by ‘are’ as this is a statement made by other authors.
L191. Replace ‘can improve’ by ‘improved’.
L195. Delete ‘when they were’.
Figure 2. Authors should explain in more detail the statistical analysis was carried out to assess oil loss: were authors comparing samples on the same day? or all together? or similar preparation during storage? As it is currently presented in the figure the statistical results (pair wise comparison) do not make sense and is very difficult to interpret.
Figure caption should contain a description of the nomenclature of the samples as you did in Figure 1.
Could the Y axis of Figure 2A be reduce to 25%.
L212. Nomenclature of the samples differ through the manuscript. Lowercase letters are used in the first half of the manuscript and capital letters from here. Please, select one way of naming the samples and be consistent.
L223. Replace ‘are’ with ‘were’.
L237. Replace ‘can be seen’ with ‘were observed’
L297. Replace ‘have’ with ‘had’
Author Response
This manuscript describes the effect of glycosylation of pea protein isolate with maltodextrin in their technological properties, such as emulsifying and textural properties to be used in HIPEs. The experimental design of the project is robust, and the authors describe in great detail their methods. Further work on the statistical analysis of the results could be done to obtain a better understanding of the effect of each variable studied. The discussion of the results increase our understanding of the potential of this glycosylated proteins and their applications in food products. Authors are encourage to do some revision of the work proposed in the lines below:
L20. ‘..good structure…’ is a very subjective way to describe a property. Authors should describe this quality in scientific terms.
Thank you for your suggestion. It has been replaced (line 20)
L21. Replace ‘… more stable emulsions…’ with ‘showed that emulsions were more stable when…’
It has been corrected in the revised version of the manuscript (line 21).
L30. Authors suggest that safety is one of the main concern of consumers that demand alternative protein sources. Authors should support this statement with some figures and references.
The reviewer is right. Two references have been added to the revised version of the manuscript (line 32). The references are:
Hadi, J., & Brightwell, G. (2021). Safety of Alternative Proteins: Technological, environmental and regulatory aspects of cultured meat, plant-based meat, insect protein and single-cell protein. Foods, 10(6), 1226.
Burger, T.G. et al. (2019). Recent progress in the utilization of pea protein as an emulsifier for food applications, Trends Food Sci. Technol. 86, 25–33.
L32. Replace ‘food systems’ with ‘food products’.
Done. It has been corrected in the revised version of the manuscript (line 33).
L32. When mentioning ‘adverse impact’ what type of impact are authors referring to? Please describe or specify.
We are referring to environmental impact. It has been added in the revised version of the manuscript (line 34).
L35. Replace ‘with’ with ‘due to’.
Done (line 36).
L67-68. This sentence contradicts the statement in lines 56-58. Please, rewrite in a consistent way.
Thanks for noticing. The contradiction has been solved and the correct sentence has been moved to the final part of the introduction (lines 105 and 106).
L78. This paragraph could be a continuation of the previous one.
The reviewer is right. It has been corrected in the revised version of the manuscript.
L103. Did the authors record/control the temperature of the solutions during reaction time? Was the temperature reach at two different times different? Could this have an impact on the type of reactions and chemical pathways dominating the reactions?
Yes, the temperature was recorded and controlled during reaction time. The temperature among samples was identical (90±1ºC; line 127). The type of reactions and chemical pathways are very influenced by the time and temperature placed in the boiling water, that’s why the authors always controlled the temperature during the wet glycosylation.
L104-105. How was the formation of browning/melanoidins controlled and checked? Were colour changes measured? Or melanoidins components analysed in the final products?
The authors did a wide screening of times of glycosylation (data not shown) before selecting the final glycosylation times. The assessment of browning was visually made.
L119-L157 Through the methods section it is not clear how many times the different samples were prepared, how many replicates per sample was produce and how many analytical replicates were run per analysis. Please, be more specific in each of the material and methods sections.
This information has been presented in lines 142-143, 153,162, 174-175.
L140. Stable Micro Systems are based at Surrey.
The reviewer is right. It has been corrected in the revised version of the manuscript (lines 166 and 172).
L158-161. A better description of the statistical analysis done to assess each property analysed should be done. It is not clear if a one-way anova was performed for all the properties, or if for some of them results were compared by day, or by temperature. Authors should reconsider the statistical analysis performed as several variables (time, temperature, concentration M (0, a and b)) were part of the experimental design and their individual effect and/or interactions should be studied.
One-way ANOVA was performed (it has been added in line 185). Each sample was considered individually, without taking into consideration the two different factors: time of reaction and ratio of PPI:MD. This is so because the degree of glycosylation cannot be considered for samples 15p and 30p as none of them has glycosylated hydroxyls and their behavior are only a consequence of time of heating (no reaction has occurred).
In the case of back extrusion results, two different one-way ANOVAs were performed: one of them comparing among the 6 samples, and another one comparing the time of storage.
L181-182. Rewrite this sentence. Authors should explain in more detail how non-polar amino groups became polar.
Thank you for your comment. The sentence was a bit confusing in the previous version. It has been rewritten (lines 208-209).
L183. Replace ‘were’ by ‘are’ as this is a statement made by other authors.
Done (line 2010).
L191. Replace ‘can improve’ by ‘improved’.
Done (line 218).
L195. Delete ‘when they were’.
Done.
Figure 2. Authors should explain in more detail the statistical analysis was carried out to assess oil loss: were authors comparing samples on the same day? or all together? or similar preparation during storage? As it is currently presented in the figure the statistical results (pair wise comparison) do not make sense and is very difficult to interpret.
We agree with the reviewer, it was hard to understand in the previous version of the manuscript. Figure 2A stands for storage at 4ºC and 2B for storage at 20ºC.
The authors compared among samples for the same day of storage. This has been added in lines 232-233.
Figure caption should contain a description of the nomenclature of the samples as you did in Figure 1.
The description has been included in the figure caption.
Could the Y axis of Figure 2A be reduce to 25%.
The authors have now changed the Y axis to a limit of 20% as requested by reviewer 2.
L212. Nomenclature of the samples differ through the manuscript. Lowercase letters are used in the first half of the manuscript and capital letters from here. Please, select one way of naming the samples and be consistent.
Lowercase letters are used to name the conjugates and capital letters are used to name the emulsions. This is presented in lines 132-133 and 142-143.
L223. Replace ‘are’ with ‘were’.
Done (line 255).
L237. Replace ‘can be seen’ with ‘were observed’
Done (line 270).
L297. Replace ‘have’ with ‘had’
Done (line 332).
Reviewer 2 Report
The current manuscript by Morell et al. represents the experimental glycosylation of pea protein via maltodextrin in two different ratios, and the consequent use of the modified protein to prepare 6 sunflower oil emulsions. The emulsions are later centrifugated to study their coalescence, and their texture was measured. As a conclusion, the authors determine that maltodextrin-modified protein is a better emulsion stabilizer than the protein itself. I recommend that the manuscript is returned for a major revision due to the following issues:
Major issues:
1. Similar results have been obtained by other authors for glycolyzed pea proteins before [see Refs. 1-2, which are not an exhaustive list], so the current conclusion is certainly not novel (apart from the maltodextrin). Isn’t the result expected apriori?
[1] X. Chen, Y. Dai, Z. Huang, L. Zhao, J. Du, W. Li, D. Yu, Effect of ultrasound on the glycosylation reaction of pea protein isolate–arabinose: Structure and emulsifying properties, Ultrason. Sonochem. 89 (2022) 106157. https://doi.org/10.1016/j.ultsonch.2022.106157.
[2] T.G. Burger, Y. Zhang, Recent progress in the utilization of pea protein as an emulsifier for food applications, Trends Food Sci. Technol. 86 (2019) 25–33. https://doi.org/10.1016/j.tifs.2019.02.007.
2. The methods and the experimental results are poorly depicted. They need to be illustrated better to improve the publication readability and the understanding of the experiments:
- All procedures should include the description of the parameters – what are the m1, m2 and m0 parameters in equation 1?
- It is currently implied that you measured the stability against coalescence for aging emulsions. However, it is not clear how you make the experiment – do you load a new (aged) emulsion each time or do you use the same emulsion that you centrifugated the last week and if so, did you measure a fresh emulsion that was centrifugated the same total amount of time (i.e. 2h)?
- Figure 1 should be presented as solubility as a function of the maltodextrin ratio (concentration), and then presented as two lines for 15 and 30 min of treatment. This improves readability significantly.
- Figure 2A scale should be from 0 to 20% for oil loss and it has more symbols than curves. Figure 2B is unreadable with the current coding in print – you can simply say that less than 5-10 % oil is released for the high concentrations.
- Are there explanations for the coalescence with time for the optical micrographs or these are centrifugated emulsions?
Author Response
The current manuscript by Morell et al. represents the experimental glycosylation of pea protein via maltodextrin in two different ratios, and the consequent use of the modified protein to prepare 6 sunflower oil emulsions. The emulsions are later centrifugated to study their coalescence, and their texture was measured. As a conclusion, the authors determine that maltodextrin-modified protein is a better emulsion stabilizer than the protein itself. I recommend that the manuscript is returned for a major revision due to the following issues: Major issues:
- Similar results have been obtained by other authors for glycolyzed pea proteins before [see Refs. 1-2, which are not an exhaustive list], so the current conclusion is certainly not novel (apart from the maltodextrin). Isn’t the result expected apriori?
[1] X. Chen, Y. Dai, Z. Huang, L. Zhao, J. Du, W. Li, D. Yu, Effect of ultrasound on the glycosylation reaction of pea protein isolate–arabinose: Structure and emulsifying properties, Ultrason. Sonochem. 89 (2022) 106157. https://doi.org/10.1016/j.ultsonch.2022.106157.
[2] T.G. Burger, Y. Zhang, Recent progress in the utilization of pea protein as an emulsifier for food applications, Trends Food Sci. Technol. 86 (2019) 25–33. https://doi.org/10.1016/j.tifs.2019.02.007.
There are several differences that make our work quite different from the rest of publications. In our study, we aimed to investigate the functionalization via glycosylation of a commercial pea protein isolate (PPI) with maltodextrin (MD) to act as an emulsifier in high-internal-phase emulsions (HIPEs). All the works that are previously published studied low or very low oil concentration emulsions. In our samples (HIPEs) we used almost 80% of oil, and it is very difficult to emulsify this high quantity of oil. This is the first study where glycosylated pea protein is used to stabilize/emulsify HIPEs.
Moreover, to understand the relevance of glycosylation time, avoiding browning o melanoidin formation, in order to use the conjugates for industrial applications, we establish short glycosylation times, which are quite different from the previous studies published, with longer times of reaction (more than 60 min).
With the unusual and elevated oil concentration and the shorter times of reaction, the results were not expected a priori.
The introduction section has been improved (lines 69-91), and the 2 references given by the reviewer have been included in the introduction section (lines 39 and 32, respectively). Regarding these 2 references, in the first one, the authors use short times of reaction, but with ultrasound technology and very low oil concentrations (5%), which are the main differences if compared to our work. The second reference is a review with no information about the use of glycosylated pea protein for emulsifying concentrated emulsions.
- The methods and the experimental results are poorly depicted. They need to be illustrated better to improve the publication readability and the understanding of the experiments:
- All procedures should include the description of the parameters – what are the m1, m2 and m0 parameters in equation 1?
M&M section has been improved (lines 127, 132-133, 136-138, 142-143, 153, 160-162, 174-175, 185). A description of the parameters in the equation has been added, as requested by the reviewer.
- It is currently implied that you measured the stability against coalescence for aging emulsions. However, it is not clear how you make the experiment – do you load a new (aged) emulsion each time or do you use the same emulsion that you centrifugated the last week and if so, did you measure a fresh emulsion that was centrifugated the same total amount of time (i.e. 2h)?
We loaded a new (aged) emulsion each time.
- Figure 1 should be presented as solubility as a function of the maltodextrin ratio (concentration), and then presented as two lines for 15 and 30 min of treatment. This improves readability significantly.
Figure 1 has been changed to improve its readability.
- Figure 2A scale should be from 0 to 20% for oil loss and it has more symbols than curves. Figure 2B is unreadable with the current coding in print – you can simply say that less than 5-10 % oil is released for the high concentrations.
Figure 2 has been improved. In Figure 2A, the authors have changed the Y axis to a limit of 20 %. Code color has been changed and lines have been thickened in both figures for a better visualization.
- Are there explanations for the coalescence with time for the optical micrographs or these are centrifugated emulsions?
The explanations given in the discussion are for samples without being centrifuged.
Reviewer 3 Report
The introduction should be considerably improved as there are many inconsistencies. Glycosylation should be explained in more detail.
Why were the conjugates freeze-dried and not directly used to produce the HIPES? Don´t the author think that using lyophilization might limit the use of these systems on a commercial scale?
The study lacks a more profound understanding and characterization of the conjugates after the treatment. More analysis should have been carried out to understand how the treatment affected the primary and secondary structure of the conjugates. Only solubility tests were carried out, so why not analyze changes in droplet size, zeta potential, hydrophobicity, and FTIR, among others? I do not consider the analyses that were carried out are sufficient. Moreover, the article lacks more in-depth discussions, it mainly just reports their results.
Moreover, the conclusion is not insightful, what are the authors suggestions? It simply makes a summary of results that were already mentioned.
Other minor comments:
Line 12: the use
Line 29: rephase
Line 32: Adverse impact in what? This should be mentioned.
Line36: Are they abundant?
Line 43-44: Is there really a lack of knowledge about its techno-functional in pea protein?
Line 46: This is not necessarily the case, there are many studies comparing pea protein with protein in varying concentrations of protein, protein-polysaccharide ratios.
Linee77: What is meant by “ultra” stable?
Line 124: Why was solubility measured using in phosphate- buffered saline? Weren’t the HIPEs produced in distilled water? Why not carry out solubility measurements also in distilled water?
Line 299: Why weren’t these analyses carried out so that author could affirm that for sure?
Author Response
The introduction should be considerably improved as there are many inconsistencies. Glycosylation should be explained in more detail.
Inconsistencies have been addressed. Moreover, the introduction has been considerably improved and glycosylation has been explained with more detail.
Why were the conjugates freeze-dried and not directly used to produce the HIPES? Don´t the author think that using lyophilization might limit the use of these systems on a commercial scale?
We agree to the reviewer; lyophilization can limit the commercial use of the conjugates. We used it to stabilize the conjugates in order to be used for the preparation of the HIPEs replications, because we wanted to use the same conjugate for the three replications, which were carried out in different dates. Other authors as Wen et al. (reference [32] included in the manuscript, line 120) also freeze- dried the conjugates after their obtention.
The study lacks a more profound understanding and characterization of the conjugates after the treatment. More analysis should have been carried out to understand how the treatment affected the primary and secondary structure of the conjugates. Only solubility tests were carried out, so why not analyze changes in droplet size, zeta potential, hydrophobicity, and FTIR, among others? I do not consider the analyses that were carried out are sufficient. Moreover, the article lacks more in-depth discussions, it mainly just reports their results.
Moreover, the conclusion is not insightful, what are the authors suggestions? It simply makes a summary of results that were already mentioned.
The reviewer is right, a more profound analysis of the conjugates could have been carried out; however, our main objective was to analyze if the conjugates obtained at such low reaction times were useful to stabilize emulsions with a very high amount of oil (around 80%).
Regarding, the discussion of the results, they have been explained in more detail (lines 208-209, 232-233, 243-248, 261-263, 302-305), and the conclusion has been rewritten avoiding repetition of the results and including recommendations and suggestions (lines 350-359).
Other minor comments:
Line 12: the use
Done.
Line 29: rephase
The sentence has been rephrased.
Line 32: Adverse impact in what? This should be mentioned.
It has been mentioned in line 34.
Line36: Are they abundant?
Goldstein & Reifen (2022) stated that “legume proteins are abundant”. Also, Zhang, Wang & Adhikari (2022) take an idea from Can Karaka, Low and Nickkerson (2015) that “legume-based protein is considered as one of the most promising alternatives to animal protein due to its abundance and relatively low cost”. To support the idea, these references have been added to the new version of the manuscript in line 37.
Line 43-44: Is there really a lack of knowledge about its techno-functional in pea protein?
There is not really a lack of knowledge per se. The sentence has been modified (lines 44-45).
Line 46: This is not necessarily the case, there are many studies comparing pea protein with protein in varying concentrations of protein, protein-polysaccharide ratios.
This idea has been rewritten to avoid misunderstandings in the new version of the manuscript (lines 47 and 48).
Line 77: What is meant by “ultra” stable?
It has been deleted in the new version of the manuscript.
Line 124: Why was solubility measured using in phosphate- buffered saline? Weren’t the HIPEs produced in distilled water? Why not carry out solubility measurements also in distilled water?
The protein content of the supernatant after centrifugation was determined according to the Lowry method using bovine serum albumin (BSA) as the standard. Our commercial BSA recommended the dilution of the protein in phosphate-buffered saline. Therefore, we decided to follow the company recommendations in order to obtain the most accurate results.
Line 299: Why weren’t these analyses carried out so that author could affirm that for sure?
As stated before the main focus of our work were the concentrated emulsions. Nevertheless, the sentence has been changed (line 332-336) for a better understanding.
Round 2
Reviewer 1 Report
The authors have addressed all the reviewers' comments in a very thoughtful way improving substantially the quality of the manuscript.
Author Response
Thank you very much for your help
Reviewer 2 Report
The revised version of the manuscript presents an improved description of the experimental procedures and the results obtained. Figures are now easier to read, and the equations are described as they are supposed to be. The authors have also included additional text to illustrate the importance of their study in the introduction (here I recommend that they include the text, used in their response also for better emphasis).
The study is now almost complete, as it only lacks an explanation for the observed coalescence phenomenon, which is confirmed both visually and implicitly through the texture analyzer. As the emulsions increase their sizes more than twice over a period of 4 weeks, it becomes obvious that the initial drop size obtained by the authors is probably due to changes in the viscosity of the continuous phase (increases with protein modification), and the subsequent growth is due to slow coalescence until a dense adsorption layer of the modified protein is obtained. I recommend that the authors calculate what is the adsorption of the protein at equilibrium and estimate if they are in a “surfactant-poor” regime of emulsification at this high protein concentration (e.g. https://doi.org/10.1039/B715933C) or if there is another explanation for the coalescence. Once this check is performed, I would recommend the manuscript for publication.
Author Response
We want to thank the reviewer for the suggestions; they have allowed improving our manuscript a lot.
You have our responses below:
The revised version of the manuscript presents an improved description of the experimental procedures and the results obtained. Figures are now easier to read, and the equations are described as they are supposed to be. The authors have also included additional text to illustrate the importance of their study in the introduction (here I recommend that they include the text, used in their response also for better emphasis).
We have added these ideas in lines 88-93 and 107- 117
The study is now almost complete, as it only lacks an explanation for the observed coalescence phenomenon, which is confirmed both visually and implicitly through the texture analyzer. As the emulsions increase their sizes more than twice over a period of 4 weeks, it becomes obvious that the initial drop size obtained by the authors is probably due to changes in the viscosity of the continuous phase (increases with protein modification), and the subsequent growth is due to slow coalescence until a dense adsorption layer of the modified protein is obtained. I recommend that the authors calculate what is the adsorption of the protein at equilibrium and estimate if they are in a “surfactant-poor” regime of emulsification at this high protein concentration (e.g. https://doi.org/10.1039/B715933C) or if there is another explanation for the coalescence. Once this check is performed, I would recommend the manuscript for publication.
We have improved the explanation of the coalescence mechanism based on the reference given by the reviewer, which has been included in the manuscript (lines 348-353).